# Antibiotic perceptions, adherence, and disposal practices among parents of pediatric patients

Keerti L. Dantuluri[1]*, Kemberlee R. Bonnet[2], David G. Schlundt[2], Rebecca J. Schulte[2], Hannah G. Griffith[3], Alexandria Luu[3], Cara Charnogursky[4], Jessica M. Perkins[5], Christine C. Whitmore[3], Ritu Banerjee[4], Leigh M. Howard[4], Carlos G. Grijalva[3]

1 Division of Infectious Diseases, Department of Pediatrics, Levine Children's Hospital at Atrium Health, Charlotte, North Carolina, United States of America, 2 Department of Psychology, Vanderbilt University, Nashville, Tennessee, United States of America, 3 Department of Health Policy, Vanderbilt University Medical Center, Nashville, Tennessee, United States of America, 4 Division of Infectious Diseases, Department of Pediatrics, Vanderbilt University Medical Center, Nashville, Tennessee, United States of America, 5 Department of Human and Organizational Development, Peabody College, Vanderbilt University, Nashville, Tennessee, United States of America

☯ These authors contributed equally to this work.
* keerti.dantuluri@atriumhealth.org

**Data Availability Statement:** All relevant data are within the manuscript and its Supporting information files.

## Abstract

Antibiotics are frequently prescribed for children in the outpatient setting. Although sometimes necessary, antibiotic use is associated with important downstream effects including the development of antimicrobial resistance among human and environmental microorganisms. Current outpatient stewardship efforts focus on guiding appropriate antibiotic prescribing practices among providers, but little is known about parents' understanding of antibiotics and appropriate disposal of leftover antibiotics. To help bridge this gap, we conducted a qualitative study to assess parental understanding of their children's antibiotics, their adherence to antibiotic instructions, and their disposal practices. We conducted a semi-structured interview with parents of 13 children diagnosed with acute respiratory illnesses and prescribed antibiotics in an urban outpatient clinic. We found that parents had limited understanding of how antibiotics work. Although they received instructions about antibiotic use during the healthcare visit, adherence to the prescription and appropriate disposal of antibiotics was suboptimal. Limited baseline understanding of antibiotics, their prior experiences with antibiotics, perceptions about their social networks' antibiotic use, and information provided to them by healthcare providers may influence these behaviors. Our findings can inform educational efforts of outpatient stewardship programs to help optimize parental understanding of how to use and dispose of their children's antibiotics.

## Introduction

Antibiotics are frequently prescribed for children with over 20% of outpatient pediatric visits resulting in at least 1 antibiotic prescription [1]. Most antibiotic prescriptions among children

**Funding:** KLD was supported by a NIAID grant T32A1095202 (PI: Mark Denison) and the Vanderbilt University Trans-Institutional Programs (TIPS) Vanderbilt Study of Antimicrobial Resistance (V-StAR) (Co-PIs: Leigh Howard and Carlos Grijalva). CGG was supported in part by NIAID grant 1K24AI148459-01. LMH was supported in part by NIAID grant K23AI141621. JMP acknowledges salary support from NIH K01MH115811. The funders had no role in study design, data collection and analysis, decision to publish, or preparation of the manuscript.

**Competing interests:** The authors have declared that no competing interests exist.

are for acute respiratory illnesses [1]. Children may be at high risk for colonization or infection with antibiotic-resistant bacteria resistant due to early initiation of recurrent antibiotic use. This pattern of use contributes to the development of antimicrobial resistance [2]. Several interconnected processes influence antibiotic use and emergence of resistant bacteria, including provider prescribing, patient adherence, and antibiotic disposal behaviors.

First, providers should prescribe antibiotics appropriately to both optimize treatment of infections and reduce further emergence of resistant bacteria. This process involves providers effectively communicating to parents how they should use antibiotics with their children. The extent to which parents understand providers' instructions is unknown.

Second, patients should provide prescribed antibiotics to their children correctly. For the pediatric population, parents' knowledge of and perceptions about antibiotics may influence this adherence process. Existing studies reveal mixed parental perspectives about antibiotics and their use among children [3–6].

Third, leftover antibiotics should be appropriately disposed. The Food and Drug Administration (FDA) offers guidelines for safe disposition of unused medications including antibiotics [7]: either drop off unused medication at a drug take-back site or mix unused medication in an unpalatable substance, place it in sealed container and discard it in the trash. However, whether providers provide clear instructions for the disposal of leftover antibiotics or how parents actually dispose of antibiotics leftover from their children is unknown. Leftover antibiotics may be stored for later use, either by the same patient or by other individuals for whom the antibiotics were not prescribed. Recent studies found that parents in the United States often kept leftover antibiotics and diverted them to others without physician consultation [8, 9]. These practices may increase the risk of contributing to antimicrobial resistance and potential adverse drug consequences. Additionally, antibiotics may also be improperly disposed into the environment. Previous assessments reported that rural water supplies have detectable antibiotic residues attributed to direct antibiotic disposal by surrounding residents [10] and reclaimed water samples used for irrigation contain low-level antibiotic concentrations even after treatment [11]. This unregulated environmental exposure has the potential to alter the resistance profiles of soil microbiota, which are an important potential source of human pathogens, and thus could substantially impact human and animal health [12].

To combat the growing threat of antimicrobial resistance, many children's hospitals and outpatient settings have implemented antibiotic stewardship programs that promote judicious use of antibiotics [13]. However, few studies have assessed parent perspectives about these processes. To address these gaps, we conducted one-on-one semi-structured interviews with parents of children who were prescribed antibiotics for acute respiratory illnesses. We queried their knowledge, perceptions, and attitudes towards antibiotics, understanding of antibiotic-associated side effects, adherence to antibiotic administration instructions, and disposal of leftover antibiotics.

## Materials and methods

### Study design and population

A team of pediatric infectious disease physicians, epidemiologists, psychologists, sociologists, and undergraduate and public health graduate students developed a semi-structured interview consisting of 25 closed-ended questions and 11 open-ended questions (S1 Fig). The questions explored parental knowledge of antibiotics, their adherence to instructions, and their disposal practices. We conducted the interview in-person to allow for authentic responses from our participants by repeatedly assuring them that there were no right or wrong answers. The

survey was pilot tested with three individuals in the non-medical community and revised itera-tively to improve clarity before conducting formal interviews.

Parents of eligible children were approached at an urban primary care pediatric outpatient clinic affiliated with an academic medical center in Davidson County, Nashville, Tennessee, United States. Children were eligible for study participation if they were ≥2 months and ≤5 years of age, diagnosed with an acute respiratory illness (ARI), and prescribed an antibiotic. We included a younger age group of children in order to capture parental perceptions of chil-dren who are commonly prescribed antibiotics. ARI diagnoses included acute otitis media, pneumonia, pharyngitis, sinusitis, bronchiolitis, and bronchitis. Children were excluded if they had a tracheostomy, received home health nursing, had a parent who was is in the medical field (physician, advance care practitioner, nurse, pharmacist), had a parent who did not speak English, or if they lived outside of Davidson County.

If a parent of an eligible child consented to the study, a home visit was scheduled 6–8 days after the enrollment visit to assess antibiotic adherence and perform the one-on-one interview. This visit was scheduled as part of another study to assess the changes in children's nasal resis-tome after a short course of antibiotics [14]. Team members involved in interviewing partici-pants and coding transcripts met periodically to review study progress and to determine, by consensus, if thematic saturation had been reached. Enrollment took place 3 half-days per week from August 2019 until March 2020, when thematic saturation, or the point where addi-tional interviews no longer generate new information, was achieved [15].

## Data collection

At the home visit, two study investigators (KLD and HGG) conducted a 15–20-minute one-on-one semi-structured interview with the participating parent. Answers to all open-ended questions were audio-recorded, and then transcribed by a professional transcription service [16]. Transcripts were de-identified and checked for accuracy. Answers to close-ended ques-tions and demographic questions were recorded into REDCap [17]. Participants were com-pensated with a $35 gift card upon completion of the visit.

## Qualitative data analysis methods

Qualitative data coding and analysis was managed by the Vanderbilt University Qualitative Research Core, led by a PhD-level psychologist. Three authors (DGS, KRB, and RJS) developed and refined a hierarchical coding system using the interview guide and a preliminary review of the first 7 transcripts [18–21]. Major categories included: 1) antibiotic knowledge and under-standing 2) benefits of antibiotics 3) concerns 4) type of antibiotic information provided by healthcare professionals; and 5) use and handling. Major categories were divided from one to seven subcategories, with each subcategory having additional hierarchical division to capture further thematic detail. Definitions were written for the use of each category.

In order to reduce coding bias, two trained coders independently coded two of the tran-scripts. Coding of each transcript was compared and any discrepancies resolved. After reach-ing consensus, one coder coded the remaining transcripts. Each statement within the transcript was treated as a separate quote and each quote could be assigned up to five different codes. Transcripts were then combined and sorted by code. The transcripts were stored using Microsoft Excel (version 2016) and the coded quotes were processed using SPSS (version 26).

We used iterative inductive and deductive approaches for the qualitative data analysis [18–20]. Inductively, we used the sorted coded quotes to generate a detailed understanding of parents' perceptions about antibiotic use. Deductively, we applied knowledge of clinical care and Social Cognitive Theory to understand adherence and the factors that influence adherence

[22–24]. Combining these findings, we developed a conceptual framework to highlight the processes and factors influencing parents' handling of antibiotics prescribed to their child. The framework development was iterative in that it was theoretically informed, while the detailed content resulted from the qualitative interview data.

### Quantitative data analysis methods

In addition to capturing the participants' open-ended survey responses in the qualitative analysis described above, we used descriptive analysis to report the results of the closed-ended questions.

### Ethics statement

The study protocol was approved by Vanderbilt University Institutional Review Board (IRB#190885). Written informed consent by signature was obtained from all participating parents before enrollment.

## Results

### Participant characteristics

Of 25 parent-child dyads who met study selection criteria, parents of 13 (52%) children agreed to enroll in the study. Six (46%) children were <12 months of age, 7 (54%) were female, and 11 (85%) were Black (Table 1). Most children had been diagnosed with acute otitis media, and all but one had been prescribed amoxicillin. Most participants were unmarried mothers with some college education (Table 1).

### Conceptual framework

The conceptual framework is depicted in Fig 1 and illustrates the multi-faceted process of antibiotic handling that a parent engages in after receiving a prescription for an antibiotic for their child diagnosed with an acute respiratory illness. We describe observations for each component of the model and report key quotes from open-ended questions for each component in S1 Table. Participant responses to closed-ended questions are included in S2 Table. We will discuss each element of the conceptual framework in more detail in the next paragraphs.

### Clinic visit

**Antibiotic beliefs.** Parents had a wide range of beliefs about what an antibiotic is and how it works (S1 Table, Quote (Q) 1–9). Some parents understood that antibiotics were indicated for treating bacterial infections saying, "I think it fights the bad bacteria." (S1 Table, Q2). Whereas others had an incomplete or complete misunderstanding of antibiotics (Q3-9) for example, "I'm guessing it helped by reducing whatever pain that was in her ear." (Q6), and "I think it just creates the antibodies in the body to fight off the infection is my understanding of how they work." (Q7).

**Expectations.** Going into the visit, parents had varying expectations about whether their child would be prescribed an antibiotic. These expectations were influenced by their child's symptoms and previous use of antibiotics for their child/children. For example, if symptoms indicated Streptococcal pharyngitis, then they expected an antibiotic (Q14). Beliefs about antibiotics also influenced expectations of effectiveness (Q14) and side effects (Q15-17) with 6 parents (46%) reporting they were concerned about potential harmful side effects of the antibiotic (S2 Table). These beliefs and expectations interacted with each other going into the visit and influenced later steps in the antibiotic process. Parents also had a range of prior

**Table 1. Sociodemographic characteristics of participants (N = 13 parent-child dyads).**

| Characteristics | Number of Participants (N = 13) N. (%) |
|---|---|
| **Children** | |
| **Age** | |
| 2–12 months | 6 (46%) |
| 12–24 months | 5 (38%) |
| 24–60 months | 2 (15%) |
| **Gender** | |
| Female | 7 (54%) |
| Male | 6 (46%) |
| **Self-reported race** | |
| Black | 11 (85%) |
| White | 2 (15%) |
| **Self-reported ethnicity** | |
| Hispanic | 1 (8%) |
| **Diagnosis** | |
| Acute otitis media | 10 (77%) |
| Preseptal cellulitis | 2 (15%) |
| Streptococcal pharyngitis | 1 (8%) |
| **Antibiotic** | |
| Amoxicillin | 12 (92%) |
| Clindamycin | 2*(15%) |
| **Parents** | |
| **Age, median (range)** | 27 (19–44) |
| **Gender** | |
| Female | 12 (92%) |
| **Self-reported race** | |
| Black | 11 (85%) |
| White | 2 (15%) |
| **Self-reported ethnicity** | |
| Hispanic | 1 (8%) |
| **Marital status** | |
| Single | 11 (84%) |
| **Number of children, mean (range)** | 2 (1–3) |
| **Highest education level** | |
| No high school degree | 1 (8%) |
| High school degree/GED | 4 (31%) |
| Some college | 7 (54%) |
| College degree | 1 (8%) |
| **Annual household income** | |
| <$25,000 | 4 (31%) |
| $25,000–$49,000 | 5 (38%) |
| %50,000–$100,000 | 3 (23%) |
| Prefer not to answer | 1 (8%) |

*1 participant was prescribed both amoxicillin and clindamycin

**Parental Antibiotic Perceptions, Adherence, and Disposal**

**Fig 1. Flowchart of parental perceptions, adherence, and disposal practices of antibiotics.** The blue boxes depict the main stages of parents receiving an antibiotic for their child. The green and yellow boxes depict factors that influence the disposition of antibiotics at each stage. The arrows pointing inward towards a box signify factors that weigh into beliefs or actions surrounding antibiotics. The arrows pointing outward signify future steps or actions impacted by previous factors.

experiences and expectations about the outcomes of treatment with an antibiotic. While many parents expected antibiotics to work well and improve their child's symptoms (S1 Table, Q10-12), some expressed thoughts that the antibiotic did not speed recovery as they expected.

## Instructions received during the clinic visit

Parents reported usually receiving detailed instructions about the antibiotic use during the visit to the healthcare provider. They recalled a physician, nurse, pharmacist, or a combination of healthcare providers providing these instructions. Three parents could name their child's prescribed antibiotic (S1 Table Q22, Q23, Q25), while others could not recall the name or were confused about the prescription (Q24, Q25). Twelve (92%) participants reported that someone from the clinic or pharmacy provided instructions for how long to give the antibiotic (e.g., 3 times a day for 10 days, until it runs out, or until they are feeling better) (S2 Table). Eight parents (64%) reported receiving instructions on how to dispose of the leftover antibiotic afterwards (S1 Table, Q30-31 and S2 Table), while others stated that disposal was not discussed (Q34).

## Treatment and adherence

Across the cohort, parents recalled several issues affecting the extent to which they followed the prescribed antibiotic treatment regimen for their child. These issues included symptom severity, dosage timing, and issues around who was responsible for giving the antibiotic to the child.

**Daily adherence.** Parents cited several reasons for poor antibiotic adherence including inconvenient timing of administration, forgetting, and a busy schedule (S1 Table, Q35–38). Close to half of parents reported significant concerns about antibiotic side effects.

**Stopping.** Despite nearly all caregivers (92%) reporting that they received providers' instructions on antibiotic use duration (S2 Table), responses to our open-ended questions indicated that many remained confused. Participants described different interpretations of when to discontinue the antibiotic, including when they were told to stop the medication, when the medication runs out, after a specific number of days, and when the child feels better (S1 Table, Q39-45).

## Disposal

Eleven (85%) parents reported that they planned to discard antibiotics, but 8 (62%) reported that they have kept their children's left-over antibiotics in the past (S2 Table).

All 13 participants (100%) said that most of their close friends or relatives had kept leftover antibiotics from their children's illnesses in their homes (S2 Table). Additionally, 6 participants (46%) thought that most of their close friends and relatives give or handover leftover antibiotics to other people.

**Keeping leftover antibiotics.** Reasons for delayed disposal of antibiotics included the potential need to reuse it for the child for the same or a new illness, sharing the antibiotics with someone else in the family, the expensive cost of antibiotics, forgetting about antibiotics, or not knowing how to dispose of the antibiotics (S1 Table, Q46–50). Parents report previously keeping leftover antibiotics anywhere from weeks to months or years or until they are expired (Q51).

**Disposal.** Disposal methods included throwing leftover antibiotics in the trash, pouring them down the drain (and then throwing away the bottle), or taking the medication back to the pharmacy (S1 Table, Q52-57). Four parents did not know the "correct" way they should be disposing the leftovers and stated they had not been told by the providers/pharmacists (S1 Table, Q58).

## Discussion

In a small-scale and in-depth assessment of parental perceptions and practices related to prescribed antibiotic use for children and associated antibiotic disposal, we observed that parents commonly experience misunderstanding surrounding how antibiotics work, how antibiotics should be used, and how leftover antibiotics should be disposed. These observations underscore fundamental challenges for efforts channeled towards appropriate use of antibiotics after prescribing.

Our findings suggest that several factors drive poor parental adherence to antibiotic prescriptions including an incomplete understanding of how antibiotics work and miscommunications between providers and parents. The findings also reflect previously published reports about parental concern for side effects for treatment regimens for their children and how that may impact adherence to antibiotic prescriptions [25]. Improving the effectiveness of providers' communication and education of parents regarding the importance of antibiotic adherence remains an important target for antibiotic stewardship efforts.

Moreover, our finding that residual antibiotics are frequently saved for future illnesses in the child, or for diversion to others for whom the prescription was not intended, is supported by the following studies finding high rates of antibiotic diversion. A prior study revealed that nearly half of 454 parents saved leftover antibiotics from a child's prescription rather than disposing of them [8]. Among these adults, nearly three-quarters reported subsequent diversion of those antibiotics to siblings, unrelated children, unrelated adults, or themselves [8]. A study conducted in 2019 among 444 caregivers of children presenting to pediatric urgent care sites in Atlanta revealed that 12% of caregivers administered a nonprescription antibiotic to their child [9]. Our study finding that all parents assume their social networks are also keeping leftover antibiotics at home builds on an extensive literature identifying social norms as strong drivers of personal behavior.

Using leftover antibiotics in unintended recipients without provider oversight may lead to significant harm including exposure to medication side effects, inappropriate dosing, allergic reactions, or untoward interactions with other medications in addition to likely use of antibiotics for inappropriate indications. We found that many parents may be unaware of these

potential harms and may believe that their social networks are using leftover antibiotics in these ways without visible consequence. One study found that social normal feedback for high antibiotic-prescribing general practitioners in England led to a decrease in the rate of individual and overall antibiotic prescriptions [26]. Perhaps a similar social norm intervention can be applied to parents of children who receive antibiotics in order to reduce the rates of inappropriate diversion of leftover antibiotics. Our findings highlight that further counseling of parents and their local communities about the risks associated with misuse of antibiotics is needed and may be another target for future antibiotic stewardship efforts.

Antibiotics that are improperly disposed of may cause harm on a larger scale due to accumulation of antibiotic-resistant bacteria or antibiotic resistance genes in the environment, which are increasingly recognized as emerging environmental pollutants [27]. Our study reveals that knowledge of proper antibiotic disposal practices is severely lacking. With only one participant correctly identifying that leftover antibiotics can be disposed of at a pharmacy drop-off location (Q57), most responses indicate a knowledge gap in the way to properly dispose of medications, as recommended by the FDA [7]. This problem may be compounded by a lack of awareness among providers or a lack of instructions provided by healthcare providers on proper antibiotic disposal practices. In our study, just over half of the caregivers (8 participants, 62%) report receiving instructions on how to dispose of their antibiotics. In a study of California pharmacies, in which a 'secret shopper' method was used to seek information on disposal of the antibiotic trimethoprim-sulfamethoxazole, fewer than half of pharmacies provided parents with correct disposal information, and only one-tenth reported take-back programs at their location [28]. Whether this is reflective of a lack of awareness on the part of healthcare providers (physicians, nurses, pharmacists), lack of infrastructure to support medication take-back programs, competing priorities in a busy clinical setting, or a combination of these factors is unknown. Previous dedicated counseling about safe disposal practices by a medical provider has been shown to significantly improve the rate of return of unused medications [29]. However, if medical providers are not well informed they will not be able to appropriately educate patients. Educational efforts regarding antibiotic disposal aimed at medical professionals may improve dissemination of information to the public. Further study of health care providers' knowledge about proper antibiotic disposal techniques is needed.

Outpatient antibiotic stewardship programs should consider as a priority understanding the different perspectives that parents may have about antibiotics in general, about their use, and about their disposal. Initiating a dialogue about these topics during the clinical encounter can help clarify potential misconceptions about when antibiotics are needed. Providing specific instructions to parents about dosing, timing, and disposal of antibiotics may greatly improve adherence. Additionally, discussing risks of antibiotic overuse may motivate parents to adhere to instructions, which should incorporate advice against sharing of unused antibiotics. Furthermore, healthcare providers may need additional education regarding FDA recommended disposal guidance of antibiotics to be able to better advise parents. Factors driving antibiotic adherence and appropriate disposal are multi-faceted, and our findings emphasize that proactive efforts to make instructions and recommendations on antibiotic prescriptions clear, practical, and feasible for parents may be crucial to decreasing behaviors that contribute to antimicrobial resistance. Future studies to enhance outpatient stewardship should focus on evaluating the availability and accessibility of medication take-back locations as well as implementation and evaluation of formal educational tools for outpatient providers to measure any associated change in parental adherence and disposal practices.

## Strengths and limitations

Our study was limited by a small and homogeneous sample size from a single study center, which limits the generalizability of the findings to other centers with different patient populations. However, the demographics of our population reflect individuals who tend to be underrepresented in research. Our results are concordant with similar studies with larger sample sizes that report poor patient adherence to prescribed medication regimens. Our study adds a unique parental perspective on antibiotic disposal practices and highlights an area for further study. As our study was restricted to children who were prescribed antibiotics, further investigation of perceptions of antibiotic use among parents of children who were not prescribed antibiotics would be informative.

## Conclusions

This study suggests that parents may engage in significant improper antibiotic use after antibiotics are prescribed for their children. Parents reported many factors influencing antibiotic use beliefs and behavior, including their understanding of antibiotics, prior experiences, and instructions provided to them about adherence and disposal. Our findings underscore opportunities to improve antibiotic adherence in the pediatric population and antibiotic disposal practices among parents. Pediatric outpatient stewardship programs should focus efforts to improve effective communication of the importance of antibiotic adherence and appropriate antibiotic disposal practices.

## Supporting information

**S1 Fig. Survey tool for semi-structured interview.** (see attached REDCap tool).
(PDF)

**S1 Table. Direct quotations from participants.** We report key quotes to illustrate each component of the framework. Each quote identifies the participant number, antibiotic prescribed, and child's ARI diagnosis.
(DOCX)

**S2 Table. Participant answers to closed-ended questions.**
(DOCX)

## Acknowledgments

We thank the Vanderbilt Pediatric Outpatient Clinic for assistance with the investigation.

## Author Contributions

**Conceptualization:** Keerti L. Dantuluri, Hannah G. Griffith, Jessica M. Perkins, Christine C. Whitmore, Leigh M. Howard, Carlos G. Grijalva.

**Data curation:** Keerti L. Dantuluri, Hannah G. Griffith, Alexandria Luu.

**Formal analysis:** Keerti L. Dantuluri, Kemberlee R. Bonnet, David G. Schlundt, Rebecca J. Schulte.

**Funding acquisition:** Keerti L. Dantuluri, Leigh M. Howard.

**Investigation:** Keerti L. Dantuluri, Hannah G. Griffith, Alexandria Luu.

**Methodology:** Keerti L. Dantuluri, Kemberlee R. Bonnet, David G. Schlundt, Rebecca J. Schulte, Christine C. Whitmore, Leigh M. Howard, Carlos G. Grijalva.

**Project administration:** Keerti L. Dantuluri.

**Resources:** Keerti L. Dantuluri, Kemberlee R. Bonnet.

**Software:** Kemberlee R. Bonnet, David G. Schlundt, Rebecca J. Schulte.

**Supervision:** Keerti L. Dantuluri, Leigh M. Howard, Carlos G. Grijalva.

**Writing – original draft:** Keerti L. Dantuluri, Jessica M. Perkins, Leigh M. Howard, Carlos G. Grijalva.

**Writing – review & editing:** Keerti L. Dantuluri, Kemberlee R. Bonnet, David G. Schlundt, Hannah G. Griffith, Alexandria Luu, Cara Charnogursky, Jessica M. Perkins, Christine C. Whitmore, Ritu Banerjee, Leigh M. Howard, Carlos G. Grijalva.

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
