## [Decision Letter · Decision Letter 0]

6 Dec 2022

PONE-D-22-30835Antibiotic perceptions, adherence, and disposal practices among parents of pediatric patients: a mixed-methods studyPLOS ONE

Dear Dr. Dantuluri,

Thank you for submitting your manuscript to PLOS ONE. After careful consideration, we feel that it has merit but does not fully meet PLOS ONE’s publication criteria as it currently stands. Therefore, we invite you to submit a revised version of the manuscript that addresses the points raised during the review process. If you feel any points raised by the reviewers may not be relevant to your manuscript do explain with your reasons.  Kindly use the COREQ checklist for reporting the qualitative findings. You can show how you have met or not met the different parameters in a separate table.  

We look forward to receiving your revised manuscript.

Kind regards,

Pathiyil Ravi Shankar

Academic Editor

PLOS ONE

Journal Requirements:

"Dr. Dantuluri was supported by a NIAID grant T32A1095202 (PI: Mark Denison) and the Vanderbilt University Trans-Institutional Programs (TIPS) Vanderbilt Study of Antimicrobial Resistance (V-StAR) (Co-PIs: Leigh Howard and Carlos Grijalva). Dr. Grijalva was supported in part by NIAID grant 1K24AI148459-01. Dr. Howard was supported in part by NIAID grant K23AI141621. JMP acknowledges salary support from NIH K01MH115811. We thank the Vanderbilt Pediatric Outpatient Clinic for assistance with the investigation."

"KLD was supported by a NIAID grant T32A1095202 (PI: Mark Denison) and the Vanderbilt University Trans-Institutional Programs (TIPS) Vanderbilt Study of Antimicrobial Resistance (V-StAR) (Co-PIs: Leigh Howard and Carlos Grijalva). CGG was supported in part by NIAID grant 1K24AI148459-01. LMH was supported in part by NIAID grant K23AI141621. JMP acknowledges salary support from NIH K01MH115811. The funders had no role in study design, data collection and analysis, decision to publish, or preparation of the manuscript."

Reviewers' comments:

Reviewer's Responses to Questions

**Comments to the Author**

1. Is the manuscript technically sound, and do the data support the conclusions?

Reviewer #1: Partly

Reviewer #2: Yes

Reviewer #3: Yes

Reviewer #4: No

Reviewer #5: Yes

2. Has the statistical analysis been performed appropriately and rigorously? 

Reviewer #1: No

Reviewer #2: Yes

Reviewer #3: N/A

Reviewer #4: No

Reviewer #5: Yes

3. Have the authors made all data underlying the findings in their manuscript fully available?

Reviewer #1: Yes

Reviewer #2: Yes

Reviewer #3: Yes

Reviewer #4: Yes

Reviewer #5: Yes

4. Is the manuscript presented in an intelligible fashion and written in standard English?

Reviewer #1: Yes

Reviewer #2: Yes

Reviewer #3: Yes

Reviewer #4: No

Reviewer #5: Yes

5. Review Comments to the Author

Reviewer #1: This study is relevant to the current clinical scenario. However the sample size is too small to derive any statistical output and make conclusion. There are some challenges which are noticed in the manuscript which may require serious attention:

1.There are minor grammatical error at the following lines: 52,53,70,74

2. The sample size is too small to make any conclusion . This makes your research output weak and non conclusive as your data also includes quantitative data not only qualitative data. You could look at increasing your sample size for quantitative data.

3. In the demographic details there is no mention of males patients

4. The qualitative aspect of the methodology is clearly explained. However, the quantitative methodology is not clearly explained in the manuscript which need further elaboration eg: the scales used for the questionnaire.

5. Why was the qualitative data collected via interview and not self-administered considering that the respondents had basic education

6. We see that most of the respondents were unmarried women, wouldn't that be a bias and affect your results? Any reason for this finding?

Reviewer #2: This is a nice qualitative manuscript evaluating parent's perceptions about antibiotic use, adherence, and disposal practices among pediatric patients with respiratory tract infections. The authors interviewed 13 parents of young children who received antibiotics. Almost all of them were for acute otitis media. The parents provided a variety of responses about how antibiotics worked and plans for disposal. The interviews were then transcribed and coded using qualitative software. The analysis appears robust the conclusions sound. The authors did a nice literature review supporting that several of their findings were similar to surveys. The manuscript is also well written. I only have a few comments for the authors.

1) Why did you choose the ages of 2-5? I assume that this is because it was part of a larger study. If this is the case, please include that detail.

2) You only interviewed children who received antibiotics, and this was almost entirely for acute otitis media. Patients perceptions about antibiotics and how they work may be different for parents who receive antibiotics vs. those that did not. This should be considered in the limitations section.

3) I think the disposing of medications is really fascinating. The authors highlight a lot of limitations about disposing them e.g., pharmacies also giving bad advice, lack of takeback locations etc. Can the authors describe any potential pragmatic solutions for people who may want to work on this issue?

4) The demographics skew very much towards unmarried mothers with low incomes who are minorities. This should also be highlighted. I am not sure if the results would look different with other populations. If you suspect that they are the same regardless of demographics, I would also state that and describe why.

Reviewer #3: Thank you very much for the invitation to review the manuscript entitled “Antibiotic perceptions, adherence, and disposal practices among parents of pediatric patients: a mixed-methods study”. I think this is a well-conducted study and a well-written manuscript. In addition, the study addresses an important topic on the rational use of antibiotics from the patients’ perspectives. For further improvement, I have the following comments:

1. On page 4, line 65, I think the authors could edit the beginning of the 3rd paragraph of the introduction. This is because it starts with “Third, ” as a continuation of the idea addressed in the 2nd paragraph, i.e., the interconnected processes influencing antibiotic use.

2. On page 6, line 104, please spell out ARI as this is the first time mentioned in the manuscript.

3. On page 7, line 123, please add a reference to REDCap. This is because some readers might not be familiar with this tool/software, i.e., https://projectredcap.org/

4. On page 7, line 115, please explain how thematic saturation is defined in this study, and please support with (a)reference(s).

5. On page 11, line 182, please indicate that Q stands for a quote. This is to ensure this abbreviation is clear for all the readers.

6. I think the discussion could be further enhanced with more comparisons with the literature.

7. In the title, the authors stated that this is “a mixed-methods study”. In the abstract (page 3, line 35), the authors stated “we conducted a qualitative study”. In addition, the methods section is focused on the qualitative study. I feel this is mainly a qualitative study using semi-structured interviews. The quantitative part with the same 13 participants (i.e. closed-ended questions in S2 Table) is to complement the major qualitative part. Consequently, what about editing this in the title to better reflect the study design?

Thank you and all the best.

Reviewer #4: It is an important topic on Antibiotic perceptions, adherence, and disposal practices among parents of pediatric

patients.

Introduction: English: to be improved

Material and Methods:

Where the study was conducted? urban outpatient clinic? Please mention about the location/state.

NVivo software was used data analysis? As the data is mostly qualitative. Sentence no. 139 and 140 suggests that the transcripts and coded quotes were managed using Microsoft Excel (version 2016) and SPSS (version 26)?

Why the sample size was 13? How was the sample selected? Whether the interviews were conducted till data saturation?

Results

Table 1: What does 163 and 164 suggests?

Only 2 antibiotics were prescribed? Clindamycin and Amoxicillin.

Discussion

Line no. 264- Be more specific. Which study the authors are referring to?

268- It is better to avoid Another survey. Please refer to the study.

Reviewer #5: Good points

1. The introduction is appropriate.

2. The Inclusion and Exclusion criteria have been appropriately included.

3. Pilot testing was done and consent was taken .Semi structured interviews with audio recording was a good idea.

4. Quantitative data was transcribed by the professionals and this was again a good idea.

Please explain and make amends accordingly

1. Why was a gift coupon worth $35, given to the participants? This seems to be unethical.

2. Sample size seems to be too small. How did you determine the sample size and what were your limitations?

3. Please mention what additional information you could derive from the structured interviews, besides that obtained from the questionnaire. Was data saturation taken into consideration when piloting the questionnaire? Did questionnaire planning precede the interview?

4. Reference 11, Page number missing.

5. What was the need for the team members to meet when decoding the interview transcripts? Could it have given rise to some bias and how did you take care of the investigators bias.

Other Points:

1. Title page missing

2. Table caption can be added.

3. Funding acknowledgement not to be done anywhere in the manuscript .Please check if salary support is a funding acknowledgement.

7. Would you like your identity revealed to the authors of this submission?

Answer: No

8. Do you have any potentially competing interests?

Answer: "None."

9. Do you want to get recognition for this review on Publons

Answer: No

6. PLOS authors have the option to publish the peer review history of their article (what does this mean?). If published, this will include your full peer review and any attached files.

Reviewer #1: No

Reviewer #2: **Yes: **Michael Durkin

Reviewer #3: No

Reviewer #4: **Yes: **Dr. Indrajit Banerjee

Reviewer #5: No

---

## [Author Response · Author response to Decision Letter 0]

13 Jan 2023

Reponses to Editor and Journal Comments:

1. Kindly use the COREQ checklist for reporting the qualitative findings. You can show how you have met or not met the different parameters in a separate table.

Author Response: Thank you for this suggestion. We have uploaded the COREQ checklist table with corresponding page numbers that address the different parameters. 

Author Response: Thank you for this reminder. We reviewed the formatting guidelines and have ensured that our manuscript meets these guidelines. We made a slight edit to the title page (Page 1) to reflect formatting guidelines.

"Dr. Dantuluri was supported by a NIAID grant T32A1095202 (PI: Mark Denison) and the Vanderbilt University Trans-Institutional Programs (TIPS) Vanderbilt Study of Antimicrobial Resistance (V-StAR) (Co-PIs: Leigh Howard and Carlos Grijalva). Dr. Grijalva was supported in part by NIAID grant 1K24AI148459-01. Dr. Howard was supported in part by NIAID grant K23AI141621. JMP acknowledges salary support from NIH K01MH115811. We thank the Vanderbilt Pediatric Outpatient Clinic for assistance with the investigation."

"KLD was supported by a NIAID grant T32A1095202 (PI: Mark Denison) and the Vanderbilt University Trans-Institutional Programs (TIPS) Vanderbilt Study of Antimicrobial Resistance (V-StAR) (Co-PIs: Leigh Howard and Carlos Grijalva). CGG was supported in part by NIAID grant 1K24AI148459-01. LMH was supported in part by NIAID grant K23AI141621. JMP acknowledges salary support from NIH K01MH115811. The funders had no role in study design, data collection and analysis, decision to publish, or preparation of the manuscript."

Author Response: Thank you for this feedback. The current Funding Statement is accurate. We have deleted the funding information in the Acknowledgements section of the manuscript.  

Responses to Reviewer Comments:

Reviewer #1: This study is relevant to the current clinical scenario. However the sample size is too small to derive any statistical output and make conclusion. There are some challenges which are noticed in the manuscript which may require serious attention:

1. There are minor grammatical errors at the following lines: 52,53,70,74

Author Response: Thank you for pointing out these errors. We reworded what are now lines 54 – 56 to “Children may be at high risk for colonization or infection with antibiotic-resistant bacteria resistant due to early initiation of recurrent antibiotic use.” We revised line 79 to read “[…] whether providers provide clear instructions for the disposal of leftover antibiotics or how parents […],” and line 84 to read “[…] diverted them to others […].”

2. The sample size is too small to make any conclusion. This makes your research output weak and non-conclusive as your data also includes quantitative data not only qualitative data. You could look at increasing your sample size for quantitative data.

Author Response: Thank you for your comment. We agree that the small sample size is a limitation and have highlighted this point in the limitations section of the discussion (lines 388 – 391). Because we conducted in-person interviews and incorporated both open-ended and closed-ended questions in our survey, we enrolled participants until we reached thematic saturation for the purpose of our primary qualitative analysis of open-ended survey questions. Thematic saturation was achieved after we interviewed 13 participants. We cite two previously published studies (references 8 and 9) which support our findings and include larger sample sizes but are restricted to quantitative survey analyses. 

3. In the demographic details there is no mention of male patients.

Author Response: Thank you for this comment. We have added a row in Table 1 to represent male participants. 

4. The qualitative aspect of the methodology is clearly explained. However, the quantitative methodology is not clearly explained in the manuscript which need further elaboration eg: the scales used for the questionnaire.

Author Response: Thank you for requesting this clarification. We included an additional section in the methods section entitled “Quantitative data analysis methods” from lines 181 – 183: “In addition to capturing the participants’ open-ended survey responses in the qualitative analysis described above, we used descriptive analysis to report the results of the closed-ended questions.”

5. Why was the qualitative data collected via interview and not self-administered considering that the respondents had basic education

Author Response: Thank you for this question. Our goal of conducting an in-person survey was to create a comfortable and non-judgmental environment to generate authentic responses from the participants and mitigate as much as possible of the social desirability bias that occurs when participants provide responses they think are “correct.” The script for our interview (S1 Figure) reiterated multiple times that there are no right or wrong answers to the participants and we were available to clarify potential questions or concerns in a timely manner. We have added text in lines 113 – 115 to highlight this point: “We conducted the interview in-person to allow for authentic responses from our participants by repeatedly assuring them that there were no right or wrong answers.”

6. We see that most of the respondents were unmarried women, wouldn't that be a bias and affect your results? Any reason for this finding?

Author Response: Thank you for this question. We agree that this rather homogeneous population may affect the generalizability of our findings; we have addressed this concern in the discussion in lines 388 – 390: “Our study was limited by a small and homogeneous sample from a single study center, which limits the generalizability of the findings to other centers with different patient populations.” We believe that the demographics of our participants are slightly skewed because our study site was an urban primary care pediatric clinic affiliated with an academic center (lines 117 – 119) which predominantly serves children from lower socioeconomic backgrounds. However, we also consider this to be a strength of our study, as individuals who are of minority races or lower socio-economic backgrounds tend to be under-represented in research. We have added this strength in lines 390 – 391: “However, the demographics of our population reflect individuals who tend to be underrepresented in research.” 

Reviewer #2: This is a nice qualitative manuscript evaluating parent's perceptions about antibiotic use, adherence, and disposal practices among pediatric patients with respiratory tract infections. The authors interviewed 13 parents of young children who received antibiotics. Almost all of them were for acute otitis media. The parents provided a variety of responses about how antibiotics worked and plans for disposal. The interviews were then transcribed and coded using qualitative software. The analysis appears robust the conclusions sound. The authors did a nice literature review supporting that several of their findings were similar to surveys. The manuscript is also well written. I only have a few comments for the authors.

1. Why did you choose the ages of 2-5? I assume that this is because it was part of a larger study. If this is the case, please include that detail.

Author Response: Thank you for this question. We chose to only include children aged 2 months – 5 years to capture parental perceptions of antibiotics prescribed for children early in their life, as the frequency/incidence of antibiotic use in these young children is high and parental perceptions may impact both patterns of antibiotic prescribing and antibiotic administration. We excluded children less than 2 months of age as concern for bacterial infection in febrile young infants may lead to hospitalization to evaluate for sepsis. We have added text with this explanation in lines 122 – 123: “We included a younger age group of children in order to capture parental perceptions of children who are commonly prescribed antibiotics.”

2. You only interviewed children who received antibiotics, and this was almost entirely for acute otitis media. Patients perceptions about antibiotics and how they work may be different for parents who receive antibiotics vs. those that did not. This should be considered in the limitations section.

Author Response: Thank you for this important observation. We have included this limitation in lines 394 - 399: “As our study was restricted to children who were prescribed antibiotics, further investigation of perceptions of antibiotic use among parents of children who were not prescribed antibiotics would be informative.”

3. I think the disposing of medications is really fascinating. The authors highlight a lot of limitations about disposing them e.g., pharmacies also giving bad advice, lack of takeback locations etc. Can the authors describe any potential pragmatic solutions for people who may want to work on this issue?

Author Response: Thank you for the supportive comment. We agree that this is an important, yet frequently overlooked, issue. We propose that future studies should assess the feasibility and effectiveness of implementing educational activities for providers and pharmacies to educate patients and families on proper disposal of antibiotics and have highlighted this in lines 360 – 363 and 382 – 385.

4. The demographics skew very much towards unmarried mothers with low incomes who are minorities. This should also be highlighted. I am not sure if the results would look different with other populations. If you suspect that they are the same regardless of demographics, I would also state that and describe why.

Author Response: Thank you for this important comment. This issue was also raised by reviewer 1, and we addressed this point in the limitations section of the discussion in lines 388 - 391. Although this homogenous population affects the generalizability of our findings, quantitative studies with larger sample sizes and more diverse parent populations (references 8 and 9) reveal similarly high rates of antibiotic diversion (lines 303 – 305). 

Reviewer #3: Thank you very much for the invitation to review the manuscript entitled “Antibiotic perceptions, adherence, and disposal practices among parents of pediatric patients: a mixed-methods study”. I think this is a well-conducted study and a well-written manuscript. In addition, the study addresses an important topic on the rational use of antibiotics from the patients’ perspectives. For further improvement, I have the following comments:

1. On page 4, line 65, I think the authors could edit the beginning of the 3rd paragraph of the introduction. This is because it starts with “Third, ” as a continuation of the idea addressed in the 2nd paragraph, i.e., the interconnected processes influencing antibiotic use.

Author Response: Thank you for your comment. We agree that the flow of the introduction can be improved and have separated the 3 interconnected processes that influence antibiotic use into individual paragraphs. 

2. On page 6, line 104, please spell out ARI as this is the first time mentioned in the manuscript.

Author Response: Thank you for catching this error. We have spelled out ARI (acute respiratory illness) when it is first mentioned.

3. On page 7, line 123, please add a reference to REDCap. This is because some readers might not be familiar with this tool/software, i.e., https://projectredcap.org/

Author Response: Thank you for this great suggestion. We now included a reference to the website (reference 17). 

4. On page 7, line 115, please explain how thematic saturation is defined in this study, and please support with (a)reference(s).

Author Response: Thank you for this suggestion. We define thematic saturation as the point when further interviews no longer generate any new information in lines 136 - 138, and reference Green and Thorogood’s book Qualitative Methods for Health Research (reference 15). 

5. On page 11, line 182, please indicate that Q stands for a quote. This is to ensure this abbreviation is clear for all the readers.

Author Response: Thank you for catching this error. We have spelled out quote at the first occurrence of abbreviation. 

6. I think the discussion could be further enhanced with more comparisons with the literature.

Author Response: Thank you for this comment. We compared our findings to two larger quantitative studies assessing rates of antibiotic diversion among parents whose children are prescribed antibiotics in lines 303 – 322. We now expanded the discussion to include two additional previously published studies that correlate with our findings. We referenced a study (reference 25) that reports parental concern for adverse effects from atopic dermatitis treatment for their children in lines 296 - 298. We also included an example of how social norms feedback improved prescribing practices of general practitioners (reference 26) and how a similar intervention can be applied to parents of children who receive antibiotics to reduce the rates of inappropriate diversion of leftover antibiotics (reflected in lines 392 - 334).

7. In the title, the authors stated that this is “a mixed-methods study”. In the abstract (page 3, line 35), the authors stated “we conducted a qualitative study”. In addition, the methods section is focused on the qualitative study. I feel this is mainly a qualitative study using semi-structured interviews. The quantitative part with the same 13 participants (i.e. closed-ended questions in S2 Table) is to complement the major qualitative part. Consequently, what about editing this in the title to better reflect the study design?

Author Response: Thank you for your comment. We agree that this study is primarily a qualitative study. However, we do incorporate some descriptive statistical analysis of our closed-ended questions into our results and discussion, which reflects more of a quantitative analysis. To respond to Reviewer 1’s comment requesting more of a description of the quantitative component of our study, we included additional text in the methods sections in lines 180 – 183. As suggested, we also deleted “a mixed methods study” from the title. 

Reviewer #4: It is an important topic on Antibiotic perceptions, adherence, and disposal practices among parents of pediatric patients.

1. Introduction: English: to be improved

Author Response: Thank you for your comment. We have reviewed the text throughout the manuscript and revised as appropriate to attempt to ensure that the content is written clearly and is grammatically correct. Reviewer 1 also expressed concern regarding a few grammatical errors in the Introduction section and we made the following changes: We reworded lines 54 – 56 to “Children may be at high risk for colonization or infection with antibiotic-resistant bacteria resistant due to early initiation of recurrent antibiotic use.” We revised line 79 - 80 to read “[…] whether providers provide clear instructions for the disposal of leftover antibiotics or how parents […],” and line 84 to read “[…] diverted them to others […].” In response to Reviewer 3’s suggestion, we also broke up the paragraphs of the introduction to create better flow. 

2. Material and Methods: 

Where the study was conducted? urban outpatient clinic? Please mention about the location/state.

Author Response: Thank you for your question. We clarified that the study took place in an urban pediatric outpatient clinic in Davidson County, Nashville, Tennessee, USA in lines 118 – 120.

3. NVivo software was used data analysis? As the data is mostly qualitative. Sentence no. 139 and 140 suggests that the transcripts and coded quotes were managed using Microsoft Excel (version 2016) and SPSS (version 26)?

Author Response: Thank you for your question. NVivo software was not used, so we do not mention it in the manuscript. We now clarified that the transcripts were stored in Microsoft Excel, but codes were processed using SPSS in lines 162 - 164. 

4. Why the sample size was 13? How was the sample selected? Whether the interviews were conducted till data saturation?

Author Response: Thank you for your question. Although 13 is a small sample size number (which we acknowledge as a limitation in the discussion), we enrolled patients until we achieved thematic saturation. Thematic saturation was achieved after we interviewed 13 participants. In response to Reviewer 3’s suggestion, we elaborated further by defining thematic saturation as the point when further interviews no longer generate any new information in lines 137 - 138, and reference Green and Thorogood’s book Qualitative Methods for Health Research (reference 15).

5. Results

Table 1: What does 163 and 164 suggests?

Author Response: We appreciate this feedback but are unable to understand the numbers the reviewer refers to. We would request clarification of this query so that we can revise the table accordingly. 

If this question refers to the paragraph on the inductive/deductive approach to analysis, we have edited the paragraph (lines 165 - 179) lightly to be clearer about the analysis approach.

6. Only 2 antibiotics were prescribed? Clindamycin and Amoxicillin.

Author Response: Thank you for this question. That is correct, that amoxicillin and clindamycin were the only antibiotics prescribed among these 13 children. This is reflective of typical patterns of antibiotic usage for the common conditions observed, given that amoxicillin is the guideline-concordant first-line antibiotic for bacterial acute otitis media. It is also frequently used as an alternative to penicillin for Streptococcal pharyngitis given the more convenient dosing schedule. 

7. Discussion

Line no. 264- Be more specific. Which study the authors are referring to?

Author Response: Thank you for this question. We describe the two studies we are referring to (references 8 and 9) immediately following this sentence. We clarified this by rewording the phrase in what is now line 305 to “[…] the following studies [...]” to indicate that we will describe the studies.

8. 268- It is better to avoid Another survey. Please refer to the study.

Author Response: Thank you for your comment. We rephrased this line (now 309) to read “A study conducted in 2019 among 444 caregivers of children presenting to pediatric urgent care sites in Atlanta revealed that 12% of caregivers administered a nonprescription antibiotic to their child (9).” 

Reviewer #5: Good points

1. The introduction is appropriate.

2. The Inclusion and Exclusion criteria have been appropriately included.

3. Pilot testing was done and consent was taken. Semi structured interviews with audio recording was a good idea.

4. Quantitative data was transcribed by the professionals and this was again a good idea.

Please explain and make amends accordingly:

1. Why was a gift coupon worth $35, given to the participants? This seems to be unethical.

Author Response: Thank you for this question. In addition to their participation in a comprehensive survey during the initial clinic encounter, we additionally conducted home visits within a narrow visit window, which caregivers often had to adjust their schedule to accommodate. In addition, we collected nasopharyngeal samples from children at both visits. We considered that $35 was appropriate compensation for the time associated with two visits and for the collection of two respiratory samples. This incentive and its value were approved by our Institutional Review Board. 

2. Sample size seems to be too small. How did you determine the sample size and what were your limitations?

Author Response: We appreciate this point. Although 13 is a small sample size, we enrolled patients until we achieved thematic saturation. Thematic saturation was achieved after we interviewed 13 participants. Per Reviewer 3’s suggestion, we elaborated further by defining thematic saturation as the point when further interviews no longer generate any new information in lines 136 - 138, and reference Green and Thorogood’s book Qualitative Methods for Health Research (reference 15). We also expanded our discussion to reflect that the small sample size from a single study center limit the generalizability of our findings but do reflect a minority population that is often underrepresented in research. 

3. Please mention what additional information you could derive from the structured interviews, besides that obtained from the questionnaire. Was data saturation taken into consideration when piloting the questionnaire? Did questionnaire planning precede the interview?

Author Response: Thank you for your questions. We felt that we could generate more authentic responses from our participants using structured in-person interviews and mitigate as much as possible of the social desirability bias that occurs when participants provide responses they think are “correct” from a self-administered survey. For example, part of our script for the survey (S1 Fig) included frequent reminders that there are no right or wrong answers to our questions. We added text in lines 113 – 115 to highlight this point: “We conducted the interview in-person to allow for authentic responses from our participants by repeatedly assuring them that there were no right or wrong answers.” We pilot tested the survey on 3 individuals without a medical background, however we did not anticipate nearing thematic saturation until we interviewed approximately 10 participants during the study (and confirmed thematic saturation by the 13th interview). Questionnaire planning took several weeks to ensure that the viewpoints of our multi-disciplinary team were incorporated and that clarity for the general public was ascertained. 

4. Reference 11, Page number missing.

Author Response: Thank you for your comment we have edited Reference 11 to read: “Kulkarni P, Olson ND, Raspanti GA, Rosenberg Goldstein RE, Gibbs SG, Sapkota A, et al. Antibiotic Concentrations Decrease during Wastewater Treatment but Persist at Low Levels in Reclaimed Water. Int J Environ Res Public Health. 2017 Jun 21;14(6):668.”

5. What was the need for the team members to meet when decoding the interview transcripts? Could it have given rise to some bias and how did you take care of the investigators bias.

Author Response: Thank you for this question. Two coders independently coded the transcript in order to reduce coding bias. The meeting was set up to ensure there was consensus prior to a single coder coding the remaining transcripts also allowed us to reduce a single-coder bias in the coding process. We have clarified this in lines 158 - 159 to highlight the independent coding: “In order to reduce coding bias, two trained coders independently coded two of the transcripts.”

6. Title page missing

Author Response: Thank you for this comment. We have included the title page in Page 1 of the manuscript. We reviewed the formatting guidelines made a slight edit to the title page to reflect formatting guidelines.

7. Table caption can be added.

Author Response: Thank you for this comment. We have moved the table caption to the same page as the table for clarity. 

8. Funding acknowledgement not to be done anywhere in the manuscript. Please check if salary support is a funding acknowledgement.

Author Response: Thank you for this comment. The journal made the same comment and we have removed salary and project funding from the acknowledgement section.

---

## [Decision Letter · Decision Letter 1]

30 Jan 2023

Antibiotic perceptions, adherence, and disposal practices among parents of pediatric patients

PONE-D-22-30835R1

Dear Dr. Dantuluri,

We’re pleased to inform you that your manuscript has been judged scientifically suitable for publication and will be formally accepted for publication once it meets all outstanding technical requirements.

Kind regards,

Pathiyil Ravi Shankar

Academic Editor

PLOS ONE

Additional Editor Comments (optional):

Reviewers' comments:

Reviewer's Responses to Questions

**Comments to the Author**

1. If the authors have adequately addressed your comments raised in a previous round of review and you feel that this manuscript is now acceptable for publication, you may indicate that here to bypass the “Comments to the Author” section, enter your conflict of interest statement in the “Confidential to Editor” section, and submit your "Accept" recommendation.

Reviewer #1: All comments have been addressed

Reviewer #2: All comments have been addressed

Reviewer #3: All comments have been addressed

Reviewer #5: All comments have been addressed

2. Is the manuscript technically sound, and do the data support the conclusions?

Reviewer #1: Yes

Reviewer #2: Yes

Reviewer #3: Yes

Reviewer #5: Yes

3. Has the statistical analysis been performed appropriately and rigorously? 

Reviewer #1: Yes

Reviewer #2: Yes

Reviewer #3: Yes

Reviewer #5: Yes

4. Have the authors made all data underlying the findings in their manuscript fully available?

Reviewer #1: Yes

Reviewer #2: Yes

Reviewer #3: Yes

Reviewer #5: Yes

5. Is the manuscript presented in an intelligible fashion and written in standard English?

Reviewer #1: Yes

Reviewer #2: Yes

Reviewer #3: Yes

Reviewer #5: Yes

6. Review Comments to the Author

Reviewer #1: There are no more suggestions to be made .All the suggestions made to improve the manuscript has been addressed. All the best.

Reviewer #2: (No Response)

Reviewer #3: Thank you very much for addressing my comments on the earlier version. I think the revised version is now OK. I wish you all the best.

Reviewer #5: There wee a few queries that I had raised earlier. The authors have answered the queries satisfactorily.The paper may be accepted for publication.

7. PLOS authors have the option to publish the peer review history of their article (what does this mean?). If published, this will include your full peer review and any attached files.

Reviewer #1: No

Reviewer #2: **Yes: **Michael Durkin

Reviewer #3: No

Reviewer #5: **Yes: **Dr Juhi Kalra

---

## [Editor Report · Acceptance letter]

1 Feb 2023

PONE-D-22-30835R1 

Antibiotic perceptions, adherence, and disposal practices among parents of pediatric patients 

Dear Dr. Dantuluri:

I'm pleased to inform you that your manuscript has been deemed suitable for publication in PLOS ONE. Congratulations! Your manuscript is now with our production department. 

Kind regards, 

on behalf of

Dr. Pathiyil Ravi Shankar 

Academic Editor

PLOS ONE